# Neural Interruption by Unilateral Labyrinthectomy Biases the Directional Preference of Otolith-Related Vestibular Neurons

**DOI:** 10.3390/brainsci11080987

**Published:** 2021-07-26

**Authors:** Nguyen Nguyen, Kyu-Sung Kim, Gyutae Kim

**Affiliations:** 1Research Institute for Aerospace Medicine, Inha University, Incheon 22332, Korea; dr.nguyen007@gmail.com (N.N.); stedman@inha.ac.kr (K.-S.K.); 2Department of Otorhinolaryngology Head & Neck Surgery, Inha University Hospital, Incheon 22332, Korea

**Keywords:** directional preference, vestibular nucleus, otolith-related vestibular neuron, unilateral labyrinthectomy

## Abstract

Background: The directional preference of otolith-related vestibular neurons elucidates the neuroanatomical link of labyrinths, but few direct experimental data have been provided. Methods: The directional preference of otolith-related vestibular neurons was measured in the vestibular nucleus using chemically induced unilateral labyrinthectomy (UL). For the model evaluation, static and dynamic behavioral tests as well as a histological test were performed. Extracellular neural activity was recorded for the neuronal responses to the horizontal head rotation and the linear head translation. Results: Seventy-seven neuronal activities were recorded, and the total population was divided into three groups: left UL (20), sham (35), and right UL (22). Based on directional preference, two sub-groups were again classified as contra- and ipsi-preferred neurons. There was no significance in the number of those sub-groups (contra-, 15/35, 43%; ipsi-, 20/35, 57%) in the sham (*p* = 0.155). However, more ipsi-preferred neurons (19/22, 86%) were observed after right UL (*p* = 6.056 × 10^−5^), while left UL caused more contra-preferred neurons (13/20, 65%) (*p* = 0.058). In particular, the convergent neurons mainly led this biased difference (ipsi-, 100% after right UL and contra-, 89% after left UL) (*p* < 0.002). Conclusions: The directional preference of the neurons depended on the side of the lesion, and its dominance was mainly led by the convergent neurons.

## 1. Introduction

The vestibular nucleus (VN) is the core neural complex, receiving input primarily from the peripheral vestibular organ as well as other peripheral sensory structures, and the incoming messages are processed and converged in the VN. As two labyrinths are separately positioned at both sides of the head, the generated vestibular information from the ipsilateral labyrinth initially projects on the ipsilateral VN, and it is again transferred to the contralateral VN through the interconnected commissural pathway (CP) [1,2,3,4,5]. Thus, if a unilateral vestibular loss occurs, the VN fails to receive the incoming sensation of head movement, and it results in imbalanced vestibular functions such as balance control, head orientation, and navigation [6,7,8]. Vestibular dysfunction has been induced artificially by unilateral labyrinthectomy (UL) to assess the neural and behavioral effects of unilateral vestibular loss [9,10,11]. For the detection of head movement, three semicircular canals and two otoliths in each inner ear sense head rotation and linear translation, respectively. In particular, the otoliths generate directional signals during linear head translation, and the generated neural information supports highly functional performance [6]. The directional sensitivity of the vestibular neurons is determined by the hair cells in the canals and the otoliths. In each canal, afferent neurons have the same directional preference [12]. The morphological polarization of hair cells in a canal is triggered by the deflection of the cupula, and the direction for neural excitation is the same in all neurons of the canal. During head rotation, the inhibitory direction in the ipsilateral VN becomes excitatory information in the contralateral VN, and vice versa for the balanced neural information at the separated VNs [13]. On the other hand, the otolithic organ has a relatively complicated organization in the generation of its directional signal. Anatomically, the otolithic surface is unequally divided by striola; approximately two-thirds are the pars medialis and the rest are the pars lateralis, which have opposite responses to a linear translation [14,15,16,17]. The uneven responses by different portions of the structure affect neural information, and the dominant neural signal comes from the pars medialis [14,15].

Even though the underlying mechanism of the generation of otolithic signals has been well-described, few experimental results have shown the unbalanced distribution of the directional preference of otolith-related vestibular neurons after UL [18]. According to a previous study, the directional preference of otolith-related vestibular neurons under a normal condition was biased to contra-direction [19]. However, this result has rarely been supported by direct studies to identify the neural responding property (directional preference) during head linear translation. Furthermore, it has been unknown what kind of neural information mainly leads the directional preference of otolith-related vestibular neurons by unilateral vestibular loss. Here, the directional preference of the otolith-related vestibular neurons in VN was examined. In addition, the neuronal responses were classified depending on neural information, which originated from pure (only otolith) or convergent neurons (both otolith and canal). To emphasize the obtained results, a UL model was constructed by an intratympanic toxic injection, and it was investigated how unilateral vestibular loss affected directional preference in otolith-related vestibular neurons.

## 2. Materials and Methods

All experimental procedures and the laboratory animal care in this study were verified and approved by the Inha University Animal Ethics Committee (INHA 1801055-534). All animals (SD rats, male) were provided by an animal provision company (Oriental Bio Corp., Seoul, Korea).

### 2.1. Animal Preparation

Unilateral labyrinthectomy (UL) was performed using rodents (SD rats, male) by intratympanic injection of ferric chloride (FeCl_3_). For the injection, a syringe (0.5 mm-diameter needle) was inserted through the tympanic membrane until its tip encountered the middle ear ossicles. Once positioned, the needle was slightly pulled back, and the injection was conducted. To estimate a proper amount of the FeCl_3_ solution (mixture of 0.97 g FeCl_3_ and 1.4 mL saline), we initially used saline to assess the volume of the middle ear cavity. Animals were anesthetized by intramuscular injection of a mixed solution (1.3 mL/kg) of ketamine (1 µL/g) and xylazine (0.33 µL/g). Following anesthesia, saline was injected until it emerged back through the tympanic membrane. Once the backward flow of saline was observed, the injection was ceased, and the injected volume was measured. Determining the average amount based on multiple measurements, the same volume of FeCl_3_ solution was applied to animals for a UL model. After constructing the UL model, the animals were in a resting stage until they recovered from the anesthesia.

### 2.2. Behavioral Tests

Tests for both static and dynamic symptoms were conducted immediately after the animals were fully awakened, which was approximately 0.5–1 h after the moment of FeCl_3_ injection. In addition, each type of behavioral tests generally took an hour to complete. Thus, all behavioral tests were finalized within 2 h after the FeCl_3_ injection.

#### 2.2.1. Static Symptoms

Static symptoms were observed in the absence of head movements. Spontaneous nystagmus (SN), skewed deviation (SD), head deviation (HD), and paw distance (PW) were used as the critical indicators for unilateral vestibular dysfunction [20]. SN was rhythmical ocular movement with a rapid phase toward the opposite side of the lesion. SD was horizontal misalignment by ocular displacement, defined as the tilted degree between a horizontal line and the imaginary line by both eyes. HD was vertical misalignment of the head, defined as the tilted degree between a vertical line and the imaginary line by the head. PW was estimated by the distance between the two front or hind paws. For the PW measurement, the animal was placed in a transparent box immediately after its awakening, and the paw positions were identified for at least 30 s. Using the recorded positions, PW was calculated before and after UL.

#### 2.2.2. Dynamic Symptoms

The dynamic symptoms were examined in the animals’ free head movements. Three different behavioral responses, the rota-rod test, tail-hanging, and rotational direction, were used for identifying unilateral vestibular dysfunctions [20,21]. The rota-rod test has been commonly used to estimate an animal’s motor function by measuring the time that the animal is maintained on a rolling rod (5.9 cm/s). Using the tail-hanging test, it was observed whether the body was spun, which rarely happened before UL. In an open space, the animal’s directional movement was also examined to identify the rotational direction, depending on the side of the vestibular lesion. All behavioral tests were conducted to identify unilateral vestibular dysfunction after UL.

### 2.3. Extracellular Neural Recording

Following the behavioral tests, the animal was re-anesthetized by the same method (see Section 2.1. Animal Preparation). Once the animal was fully anesthetized, it was placed on the motorized stereotaxic apparatus (NEUROSTAR, Tubingen, Germany) to fix its head. The overall neural recording process followed our previous methodological approach [22,23,24] for approximately 3 h. In short, we surgically removed its scalp, and the superior surface of the skull was exposed. On the surface, the lambda was designated as a center, and the hole (2.0 mm diameter) for a recording electrode (5 MΩ, A-M System, Sequim, WA, US) was opened (generally, 3.0 mm posteriorly and 2.0 mm laterally away from the center). All extracellular recordings were taken from the right side of the brain. Consequently, translation toward the right was defined as the ipsilateral direction. Neuronal activities were explored by advancing the recording electrode, and they were tested by kinetic stimuli including horizontal rotation and linear translation following the x-axis (ipsi- and contra-lateral) [1,9,11,25]. Based on the responses to the kinetic stimulation, the neurons were classified as pure otoliths or convergent (otolith and canal-related) vestibular neurons. The pure otolith neurons showed a neuronal response only to the linear translation, while the convergent neurons responded to both the horizontal rotation and the linear translation. The neuronal responses to the stimulation were recorded with a sampling rate of 40 kHz in an OmniPlex D system (Plexon, TX, US) after they were amplified and filtered (bandpass 0.5–3 kHz). All the recordings were performed under the anesthetized condition.

### 2.4. Histological Assessment of the Peripheral Vestibular Structure

For histological assessment, selected animals were sacrificed after the neural recording. The whole skull was removed, and the enucleated brain was fixed in 10% formalin. Next, the specimen was washed in distilled H_2_O and stored in 20% EDTA for decalcification for about 2 weeks (at 4 °C). Every week, the EDTA solution was changed twice or up to three times. After the decalcification was complete, the specimen was again rinsed in distilled H_2_O, and it was implemented in a paraffin block. The paraffin block with the specimen was sliced 10μm thick, and the sliced tissues were dried at 60 °C for image acquisition. All images were prepared with 400x magnification (×10 optical and ×40 objective) (Olympus, Tokyo, Japan).

### 2.5. Data Analysis

Behavioral tests were composed of four types of static symptom and three types of dynamic symptom tests. For PW, the measured distances between the two paws in front and in back were executed off-line using a user code written in MATLAB (MathWorks, Natick, MA, US). The distances were presented in a bar chart to identify a possible width of two feet, and the normal width was assessed by measuring the largest and the smallest values (1–15 cm for the front and 1–10 cm for the hind). The final value was presented by a range instead of averaged values. As briefly explained in the Behavioral Tests section, SD and HD were examined using the captured images, comparing the misalignments with the horizontal and vertical lines, and they were assessed by the altered lines in the ocular and head displacements, respectively. The tilted angles between the normal and UL models indicated the misalignment by the malfunctioning vestibular system. SN was observed by recording ocular movement, and the abnormal movements, SN, were identified. Under the normal condition, ocular movement was rarely observed, but the UL models showed repeated ocular slides, which were the movements of SN. The tail hanging, the rotational motion, and the SN were confirmed based on the observations, and the rota-rod test was analyzed using the measured times (mean ± standard deviation). The rota-rod test basically measured the maintaining time of the animals on the rolling rod, using the rota-rod machine (Figure 1). Simply, the walking time on the rod was measured, and the same process was repeated 3–5 times. The average and standard deviation were computed before and after the FeCl_3_ injection to present the data (Figure 2C).

The directional preference was determined by the neuronal response to kinetic stimulation following the x-axis (passive inter-aural translation), which originated from the utricle. During the repeated right- or leftward linear translation, instantaneous firing rates (IFRs) of the neuron were examined, which were computed by the reciprocal number of inter-spike intervals (ISI). If IFRs increased as the head was linearly translated to the right direction, the neuron was identified as an ipsi-preferred unit. The contra-preferred neurons increased their IFRs as the heads were linearly translated to the left direction. In addition, the neuronal information responding to the horizontal rotation was examined to demonstrate whether the recording response originated from a pure or a convergent neuron. The modulation of IFRs was examined based on a curve-fitting, which was modeled by a simple sinusoidal equation as Equation (1):(1)$$C_{f}=\alpha \cdot \sin(2\times \pi \times f\times t+\beta )+\gamma$$
where curve fitting (*C_f_*) was a sinusoidal function with amplitude (*α*), phase (*β*), and constant (*γ*).

To assess statistical significance, a binomial cumulative distribution (BCD) function was applied. As all compared numbers fell into one of two groups, BCD tested if the different numbers in the two groups were statistically similar. Based on this test, the bias was determined between the groups with different directional preferences, or the ipsi- and the contra-preferred neurons.

## 3. Results

Forty-four SD rats (270–450 g, male) were used in this study. Fourteen animals were employed for the measurement of the middle ear cavity. According to the estimations, the overall volumes in the left and right ears were similar (mean ± standard deviation (STD): 50 μL ± 13.09 and 50.71 ± 9.61, respectively). Therefore, a sham or a UL model was constructed by the injection of approximately 50 μL saline or ferric chloride (FeCl_3_) solution, respectively. The rest of the animals (30) were divided into three groups: 11 sham, 14 right, and 5 left UL models. The behavioral tests were selectively applied in the UL models, and the number of animals for each symptom is summarized in Table 1, indicating unilateral vestibular dysfunction. The selections were made based on the clearness of the symptoms which were accessible for the measurements.

Although some UL models showed several different symptoms, there were no UL models that revealed all the symptoms. For example, a UL model (#02-1204) tilted its head (39.43 deg), and its performing time on the rota-rod dramatically decreased from 147 to 3.25 s before and after FeCl_3_ injection, respectively. However, its skew deviation and spontaneous nystagmus (SN) were not clearly observed. On the other hand, some UL models had no symptoms in both skew and head deviation, but their paw distance and tail hanging indicated that the chemical injection clearly damaged the peripheral vestibular system. Therefore, the assessed symptoms were determined when a UL model presented clear symptoms. According to this approach, Table 1 was summarized, and all UL models revealed two or more evident symptoms from the static and dynamic tests. For instance, the symptom of SN was observed in 7 right UL and 3 left UL animals, and half of the UL models (10/19, 52.63%) showed SN overall. Another symptom, such as tail-hanging, appeared in most UL models (15/19, 78.95%), showing a relatively high percentage in the population. As expected, the sham group had few symptoms related to unilateral vestibular dysfunction in the static and dynamic tests. Figure 1 illustrates the schematic overview of the experimental procedures. In short, the models were constructed by UL using FeCl_3_. Once the animals were awakened, some selected behavioral tests were applied to assess unilateral vestibular dysfunction. Following the tests, neuronal recording was performed to identify directional preference.

The relation between the degrees of head deviation (HD) and skewed deviation (SD) was presented depending on the side of the lesion (gray and black for the left and the right UL, respectively) (Figure 2A). Both symptoms were observed in most UL models (14/19, 70.68%, Table 1), and the circles indicate the ocular and head displacements from UL. In two models, no head tilt was observed after right UL despite the existence of SD, and another two models had no change in SD with the existence of HD. Based on the results, the head rolled to the right with right UL, while it rolled to the left with left UL. The total range of HD was between 6.12 and 42.07 deg (mean ± STD: 23.49 ± 10.76 deg), or specifically, 14.4–42.07 deg and 6.12–39.43 deg by the left and right UL, respectively. In addition to HD, the neck was often bent laterally after UL. SD was an effective clinical indicator of unilateral vestibular lesions, and this symptom normally appeared in an early phase of the behavioral responses. Right UL also induced ocular displacement, causing the right eye to move downward and the left eye to move in an upward direction, and opposite ocular displacement was induced by left UL. Due to the displacement, SD resulted in clockwise and counter-clockwise rotation by the right and left UL, respectively. Compared with the head tilt, SD was small (mean ± STD: 5.58 ± 2.29 deg), and there was little difference between the SD of the right and the left UL (5.76 ± 2.74 deg and 5.27 ± 1.79 deg, respectively). Other static responses were also selectively examined, and the results supported unilateral vestibular dysfunction. In particular, paw distance (PW) showed the asymmetry of limb positions after UL. In an examination using 6 selected UL models, the width between the two paws of the front or the hind legs changed, and the alteration in PW was evident on the hind side (Figure 2B). In the front paws, PW rarely changed after UL (average front PW: 1.26–14.36 cm and 1.08–13.87 cm before and after UL, respectively), while that of the hind side increased (53.9%, 7/13), generating a clear separation after UL (average hind PW: 2.52–4.67 cm and 4.53–6.54 cm before and after UL, respectively). The unilateral vestibular dysfunction from UL was further demonstrated by the dynamic behavioral responses. Using the rota-rod test, walking time on the rolling rod significantly decreased after UL (Figure 2C). The walking time before and after UL ranged from 44 to 335 s (mean ± STD: 208.5 ± 99.56) and from 1 to 77 s (13.23 ± 20.42), respectively. The tail hanging (15/19, 78.95%) and the rotational motion (12/19, 63.16%) were provoked after UL. The rotational direction in the tail hanging was counter-clockwise after right UL and clockwise after left UL. In an open space, the movement of the models showed the animals headed to the side of the lesion after UL. Using the same approach of constructing a model, damage by FeCl_3_ was measured based on different behavioral responses. However, this kind of assessment was limited to compare the degree of damage, and this mainly depended on the difference in morphological responses to the same amount of toxin. Therefore, a histological method might be a reliable examination for the degree of damage, as shown in Figure 5.

Following the model’s confirmation by the behavioral tests, neuronal recording was conducted, and a total of 77 neuronal responses originating from the otolith were recorded. Their detailed recording positions were shown in the Appendix A. Based on the IFR activity, all responses were classified into ipsi- and contra-preferred groups. As shown in the examples, the ipsi-preferred neuron increased its IFR as the head linearly translated to the ipsi (right) direction, while the contra-preferred neuron increased its IFR with the contra (left) movement of the head in both normal and UL models (Figure 3). Additional analysis were shown in the Appendix A.

The head was translated with an acceleration of ±63.54 cm/s^2^, and the positive and the negative areas represented the ipsi- and the contra-direction, respectively. Positive synchronization between head acceleration and IFR identified the ipsi-preferred neurons, and negative synchronization identified the contra-preferred neurons. All neuronal responses were classified by behavioral models as well as directional preference: 20 (contra-, 13 and ipsi-, 7) from the left UL, 35 (15 and 20) from the sham, and 22 (3 and 19) otolith-related vestibular neurons from the right UL (Table 2). As shown in the population, the overall numbers of neuronal responses from the left and the right ULs showed no difference (*p* = 0.561, binomial cumulative distribution (BCD)). An additional division was constructed for the separation of the pure neurons (responding to linear translation only) from the convergent neurons (responding to horizontal rotation and linear translation). The specific numbers of neurons for each group are summarized in Figure 4 (percentages) and Table 2 (numbers). In the left UL models, the labyrinthectomy generally induced the dominance of contra-preference by 13 (65%) neurons (*p* = 0.058, BCD), and the dominance was significantly led by the convergent neurons (8/9, 89%) (*p* = 0.002, BCD). On the other hand, the right UL dominantly generated an ipsi-preference (19/22) in the neuronal responses (*p* = 6.056 × 10^−5^, BCD), also leading the ipsi-dominance by the convergent neurons (100%). In the sham, there was little difference in the numbers of the ipsi- and the contra-preferred neurons (*p* = 0.155, BCD).

The histological tests on the peripheral vestibular area showed the damages from FeCl_3_ injection (Figure 5). The first row (A–C) shows the vestibular crista of the shams, and there were few damaged structures around the vestibular crista, while those of the labyrinthectomy models had interrupted vacuoles in the structure of the vestibular crista (marked by yellow arrows, D–F), agreeing with previous studies [26,27]. These interrupted swellings were observed around the crista of labyrinthectomy models, indicating vestibular damage from the chemical injection. 

## 4. Discussion

This study investigated directional preference in otolith-related vestibular neuronal responses before and after UL as well as under a normal condition (sham). As known, the chemical reaction between hydrogen peroxide and the ferric ion (Fe^3+^) increased hydroxyl radicals in cells [28], which induced oxidation, excitotoxicity, and cell death [29,30]. These reactions were how the injected FeCl_3_ damaged the peripheral vestibular end-organs. Otolith organs generally sense the linear acceleration of head movement, and the relevant neuronal responses show spatio–temporal characteristics. Under limited movement in 2D as done in the current study, otolith-related vestibular neuronal responses could be simply classified into four different preferences such as two lateral directions as well as anterior and posterior directions. However, the purpose of this study was to understand how the neural information of the canal and the otolith organs was combined, which has been a major limitation in vestibular studies. For this purpose, we identified two main neuronal responses in the VN, the pure otolith as well as the combined responses by the canal and the otolith, simply by categorizing the binary directional preference in the otolith-related vestibular responses. Based on the current observations, the biased directional preference was constructed depending on the side of the lesion, and the convergent neurons led the dominance in directional preference. Thus, unilateral vestibular loss by UL was the key to changing the dominance of directional preference.

### 4.1. Dominant Driving Force of Directional Preference

During the head linear translation, directional preference was primarily driven by otolith-related vestibular neurons, and its balance disintegrated after UL. Considering the altered dominance in directional preference after UL, it was also shown that directional preference was closely related to the incoming neural information from both sides of the otolith organs. According to our results, the loss of neural information from one side biased to the same direction. Under normal conditions (sham model), the distribution of the ipsi- and the contra-preferred neurons was even, and the pure neurons were the major group (*p* < 0.001, BCD). Under the same conditions, there was no significant dominance in the directional preference. After UL, however, the directional preference became biased, and the result was emphasized in the convergent neurons (Figure 4 and Table 2). After the left UL, contra-preferred was dominant, and ipsi-preferred was the main direction after the right UL (*p* < 0.004, BCD). Even though the neuronal type in the sham group indicated an opposite distribution compared with that after UL, the convergent neurons mainly governed the directional preference regardless of the side of the lesion. The same consequence was driven by all convergent neurons in the sham and UL models, and, thus, the main driving force for directional preference was induced by the information from the convergent neurons. 

Unlike our current sham data, on a neuroanatomical basis, it has been suggested that otolith neurons dominantly have a contra-preferred direction. However, few direct examinations on dominant directional preference have been performed. From the aspect of neural excitation, the right UL model lost concurrent neural information from the right labyrinth, and it received incoming stimulation only through the left side of the CP [11,19,25,31]. During the ipsilateral translation after right UL, hair cells initiated the excitation of the vestibular afferent neurons by the pars medialis on the left side, projecting on the right side of the VN through the CP [14,15]. Therefore, the neural recording on the right side of the VN showed increased neuronal activity as the head moved toward the right direction, while induced neural activity was dominant in the left VN. The serial activities of the utricular afferents were mainly governed by the hair cells in the pars medialis, which was located at the opposite side of the lesion. In addition, it was highly expected that the neural information from the pars lateralis would be suppressed because of the superiority of the pars medialis in the population, and that was why there were only several contra-preferred neurons.

### 4.2. Model Confirmation Based on Behavioral Responses and Histological Results

Chemical labyrinthectomy eliminated the hair cells, and it caused the asymmetric neural activity in the VN [21,32,33]. Furthermore, chemical labyrinthectomy has been known to be an easier and more efficient approach than the surgical method [29,30,34,35]. Due to the abnormal behavioral responses following unbalanced neural information, various behavioral tests have been adopted with no animal sacrifice for model evaluation [36]. A previous study reported that a toxic (e.g., sodium arsanilate) injection caused SN and HD [37], and an injection of streptomycin also induced SN and HD within 12 h [38,39]. Some experimental results demonstrated that an abnormal ocular symptom called ocular tilt reaction was caused by damage on the utricle [14,20,40,41]. These accumulative results suggested that head and ocular responses were the critical indicators for unilateral vestibular dysfunction, and examination of the responses was a reasonable method for model evaluation [20,34,42]. In addition, limb asymmetry after UL was also previously investigated during functional recovery and neurogenesis [34]. In this study, neural recording was followed by the model confirmation, and the behavioral tests were relevant with no animal sacrifice. Previously approved tests were included in the evaluation, and the analysis for PW was newly developed to identify the continuously changing symmetry in the limbs. The obtained results from static and dynamic tests indicated that the models reconstructed unilateral vestibular dysfunction by UL, and the assessments showed the analyzed consequences. Nevertheless, there was a limit to showing how much damage was caused by FeCl_3_ injection. According to a previous histological study, neither chemical nor surgical labyrinthectomy generated complete removal of the vestibular hair cells [36]. To resolve the uncertainty in the model generation, we performed an extra histological examination on the vestibular crista after behavioral tests (Figure 5). Through the examination, the damage to the peripheral vestibular organs was confirmed by the vacuoles around the crista, and this inference agreed with previous studies examining damage to the vestibular organs [26,27].

## 5. Conclusions

The directional preference of the otolith-related vestibular neurons in VN was investigated using chemical UL models as well as the normal condition. Our current study demonstrated that UL blocked the delivery of neural information originating in the otolith to the central area (VN), and its consequences were estimated by the skewed distribution of directional preference as well as by abnormal behaviors. As indicated in previous neuroanatomical studies, the neural link of two separated labyrinths maintained a balanced directional preference, but its interruption by UL biased the directional preference. Agreeing with the previous evidence, the directional preference of the otolith-related vestibular neurons in VN was biased to the ipsi- or the contra-direction, depending on the side of the lesion. In particular, the bias in directional preference was mainly governed by the convergent neurons, which received the neural information from the head rotation and the linear translation. Thus, unilateral vestibular loss heavily affected the convergent neurons.

## Figures and Tables

**Figure 1 brainsci-11-00987-f001:**
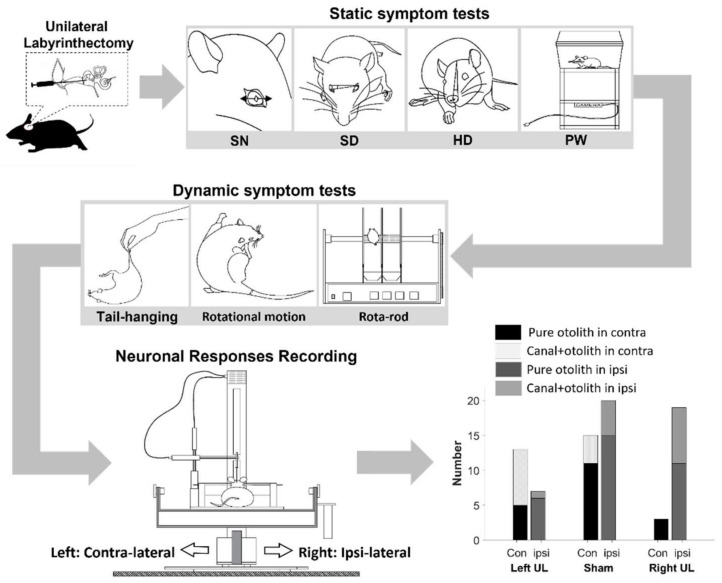
Schematic overview of experimental procedure.

**Figure 2 brainsci-11-00987-f002:**
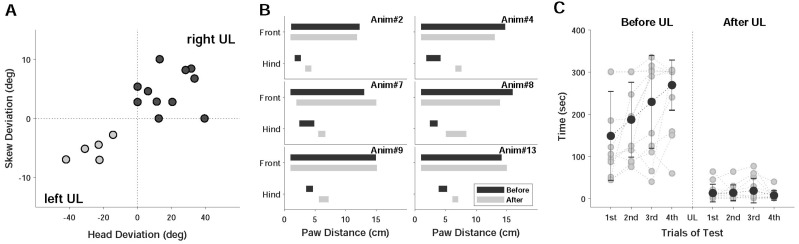
Behavioral responses before and after UL. (**A**) Relation between the degree of the head (HD) and the ocular skew deviation (SD). (**B**) Front and hind paw distances from 6 selected animals: Each bar represents the range of paw distance. (**C**) Rota-rod tests: For each condition, animals were tested 4 times.

**Figure 3 brainsci-11-00987-f003:**
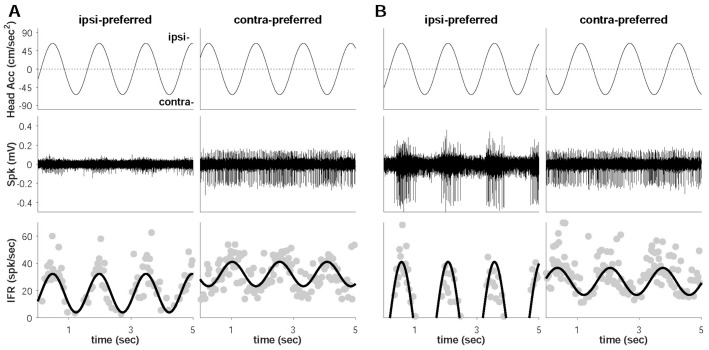
Ipsi- and contra-preferred neuronal examples responding to head acceleration. (**A**) Examples of neuronal responses from sham. (**B**) Examples of neuronal responses from labyrinthectomy model. At the bottom, the gray dots represent IFR, and the black line indicates their relevant curve-fitting based on Equation (1).

**Figure 4 brainsci-11-00987-f004:**
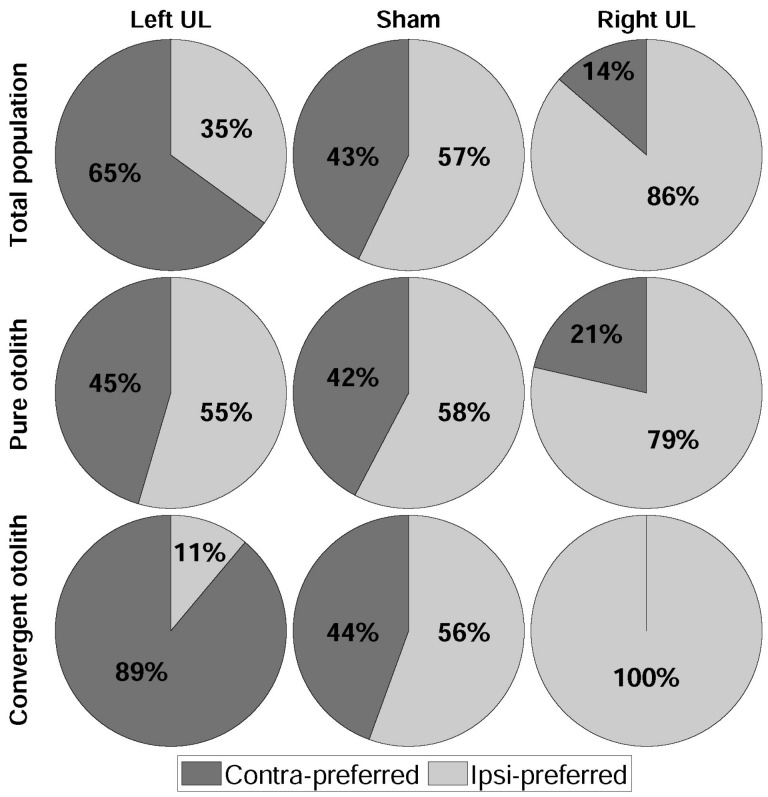
Population by lesion location. Total population in 1st row; purely otolith-related in 2nd row; otolith- and canal-related vestibular units in 3rd row.

**Figure 5 brainsci-11-00987-f005:**
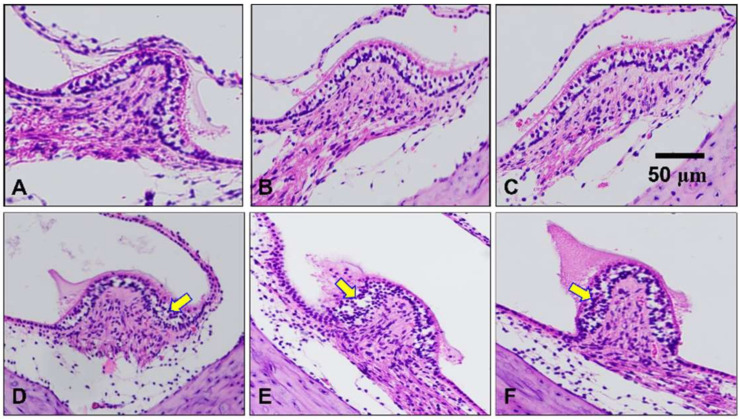
Vestibular crista from shams (**A**–**C**) and labyrinthectomy model (**D**–**F**) by FeCl_3_ injection. The damaged structures from the chemical injection are indicated by the yellow arrows.

**Table 1 brainsci-11-00987-t001:** The number of animals for each symptom in the right and left UL models.

Symptom	Right UL Model	Left UL Model	Total Number (%)
Spontaneous Nystagmus	7	3	10 (52.6)
Skew Deviation	9	5	14 (73.7)
Head Deviation	9	5	14 (73.7)
Paw Distance	8	5	13 (68.4)
Rota-Rod	7	4	11 (57.9)
Rotational Motion	8	4	12 (63.2)
Tail Hanging	13	2	15 (78.9)

**Table 2 brainsci-11-00987-t002:** The number of contra- and ipsi-preferred neurons in each group.

Symptom	Left UL ^1^		Sham		Right UL	
Preferred Direction	Cont. ^2^	Ipsi. ^3^	Cont.	Ipsi.	Cont.	Ipsi.
Pure otolith	5	6	11	15	3	11
Canal + otolith	8	1	4	5	0	8
Total	13	7	15	20	3	19

^1^ Unilateral labyrinthectomy. ^2^ Contra-preferred. ^3^ Ipsi-preferred.

## Data Availability

The data in the current study are not publicly available, but they may be available upon reasonable request.

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
