# Peer review of "Neural Interruption by Unilateral Labyrinthectomy Biases the Directional Preference of Otolith-Related Vestibular Neurons"

_brainsci, 2021, doi:10.3390/brainsci11080987_

Round 1

Reviewer 1 Report

Nicely done, no additional concerns

Author Response

Due to no comments, we have no additional responses.

Reviewer 2 Report

Considering there are very few experimental results showed the unbalanced distribution of the directional preference of otolith-related neurons after UL. Here, authors not only constructed a unilateral labyrinthectomy model with varies behavioral tests, the specific kind of neural information that dominants the directional preference of the otolith-related neurons by unilateral vestibular loss is also defined.

Major comments:

  1. What are the kinetic stimuli? And how it is performed to rats under the anesthetized state during recording? Since it is a very important part elucidate how otolith-related and convergent neuron are defined and response patterns changed under UL in contrast with sham condition as shown in Figure 3-4 and Table 2, it is essential to make the method clear enough to be convincing as well as understandable to general readers. The related citation is also absent. Meanwhile, are there any histological confirmations of extracellular recorded neurons showing they were otolith-related neurons obtained from or within the confined vestibular nuclei? The current data are not sufficient to delineate the robustness otolith-related vestibular neurons.
  2. As authors wrote in line 121-129, the linearity is key parameter to classify a pure otolith and a convergent neuron (otolith+canal-related). Is there any the functional equation or calculational method to come up with an index to show the linearity for one specific recorded neuron? Perhaps example neurons of a pure otolith and/or a convergent neuron could be provided in Figure 4?
  3. The spontaneous activities and response dynamics of the otolith-related vestibular neurons display developmental changes, which increased with age to reach a plateau at P21 for clipped cells and P14 for non-clipped cells (Lai, C. H., and Y. S. Chan. 2001. “Spontaneous Discharge and Response Characteristics of Central Otolith Neurons of Rats during Postnatal Development.” Neuroscience 103 (1): 275–88.). Beyond P14, the overall spontaneous activity increased, clipped neurons (are silenced in discharge during parts of each rotary cycle) tended to have significantly lower resting rates and higher gains than the non-clipped ones (show full-cycle responses to off-vertical axis rotation). This raises two questions:

① What is the age of animal subjects used in this study? Are they developmental mature at the same level?

② If they did, did authors also find  any clipped and non-clipped neurons among these ipsi-preferred/contra-preferred neural groups as indicated in Figure 3?

  1. In line 39-41, authors state that ‘Especially, the otoliths generate the directional signals during the linear head translation, and the generated neural information supports a highly functional performance, like navigation [6]’. While based on the work done by Ryan M. Yoder and Jeffrey S. Taube in ref. 6, and also work done by Valerio, Stephane, and Jeffrey S. Taube. (Valerio, Stephane, and Jeffrey S. Taube. 2016. “Head Direction Cell Activity Is Absent in Mice without the Horizontal Semicircular Canals.” Journal of Neuroscience 36 (3): 655–69.), the absence of normal horizontal canals results in an unstable head direction signal, and otoliths interferes with the robustness of head direction signal. It is very different from the navigation, which heavily involves hippocampal regions and spatial memory and indirectly related to projections from otolith organs and semicircular canals to vestibular nuclei (Vidal, P. P., K. Cullen, I. S. Curthoys, S. Du Lac, G. Holstein, E. Idoux, A. Lysakowski, K. Peusner, A. Sans, and P. Smith. 2015. The Vestibular System. The Rat Nervous System: Fourth Edition. Fourth Edi. Elsevier Inc.). Authors need to be careful in the introduction part when making such generalization, ‘sense of spatial orientation and self-motion’ would be more sensible than ‘navigation’ (Vidal, P. P., K. Cullen, I. S. Curthoys, S. Du Lac, G. Holstein, E. Idoux, A. Lysakowski, K. Peusner, A. Sans, and P. Smith. 2015. The Vestibular System. The Rat Nervous System: Fourth Edition. Fourth Edi. Elsevier Inc.).
  2. As authors indicated in line 62, please consider use ‘otolith-related vestibular neurons’ instead of ‘otolith-related neurons’ in the whole content, including title.
  3. In line 77-84, authors mentioned the experimental process of testing to find a proper amount of the FeCl3

① Is there an evaluation of the dose-response relationship curve for UL model in this study?

② Secondly, why did authors use the FeCl3 solution? But not others, such as, lidocaine (Magnusson, A K, and R Tham. 2003. “Vestibulo-Oculomotor Behaviour in Rats Following a Transient Unilateral Vestibular  Loss Induced by Lidocaine.” Neuroscience 120 (4): 1105–14.), bupivacaine (Zwergal, Andreas, Julia Schlichtiger, Guoming Xiong, Roswitha Beck, Lisa Günther, Roman Schniepp, Florian Schöberl, et al. 2016. “Sequential [18F]FDG ΜPET Whole-Brain Imaging of Central Vestibular Compensation: A Model of Deafferentation-Induced Brain Plasticity.” Brain Structure and Function 221 (1): 159–70.), histamine or histaminergic agonists (H3 agonist, Horii, A., Takeda, N., Matsunaga, T., Yamatodani, A., Mochizuki, T., Okakura-Mochizuki, K., et al. (1993). Effect of unilateral vestibular stimulation on histamine release from the hypothalamus of rats in vivo. Journal of Neurophysiology, 70, 1822–1826.; Takeda, N., Morita, M., Hasegawa, S., Horii, A., Kubo, T., & Matsunaga, T. (1993). Neuropharmacology of motion sickness and emesi. A review. Acta Otalaryngologica Supplement, 501, 10–15.), APV (de Waele, C., Vibert, N., Baudrimont, M., & Vidal, P. P. (1990). NMDA receptors contribute to the resting discharge of vestibular neurons in normal and hemilabyrinthectomized guinea pig. Experimental Brain Research, 81, 125–133.), muscarinic agonists (Zanni, M., Giardino, L., Toschi, L., Galetti, G., & Calza, L. (1995). Distribution of neurotransmitters, neuropeptides, and receptors in the vestibular nuclei complex of the rat: an immunocytochemical, in situ ybridization and quantitative receptor autoradiographic study. Brain Research Bulletin, 36, 443–452.; Dominguez del Toro, E., Juiz, J. M., Peng, X., Lindstrom, J., & Criado, M. (1994). Immunocytochemical localization of the alpha subunit of the nicotinic acetycholine receptor in the rat central nervous system. Journal of Comparative Neurology, 349, 325–342.)?

③ Is this directional preference of otolith-related neurons bias could only be attained using  ferric chloride intratympanic injection?

④  Could authors discuss a little more about the neural interruption caused by FeCl3 solution regarding the pharmacological interactions of vestibular neurons and their inputs in defined circuits?

  1. In section ‘2.2. Behavioral test’, could authors provide more clear details about how the static symptoms and dynamic symptoms were recorded online and analyzed off-line as shown in other works (Kim, Gyutae, Nguyen Nguyen, and Kyu Sung Kim. 2020. “Postural Control in Paw Distance after Labyrinthectomy-Induced Vestibular Imbalance.” Medical and Biological Engineering and Computing 58 (12): 3039–47. ; Tighilet, Brahim, David Péricat, Alais Frelat, Yves Cazals, Guillaume Rastoldo, Florent Boyer, Olivier Dumas, and Christian Chabbert. 2017. “Adjustment of the Dynamic Weight Distribution as a Sensitive Parameter for Diagnosis of Postural Alteration in a Rodent Model of Vestibular Deficit.” PLoS ONE 12 (11): 1–20.)?
  2. There are several citations missing in certain paragraph, such as line 54-56, line 92-94, line 104-106.
  3. Is there any clinical and functional indication of lesion-side-dependence of biased directional preference among otolith related neurons in vestibular nuclei? Which aspects could be affected more by this? Considering the vestibular nuclei receive direct input from multiple brain areas including but no limiting to the vestibular afferents and actively participate in many essential brain functions such as gaze control, posture and balance control, multi-sensory integration and so on.
  4. There many sentences are both obscure and vague. Authors need to revise the manuscript under the help of native speakers to fix this issue.

Minor comments:

  1. Did authors observe any recovery of static symptoms?
  2. In line 157, IFR was not explained until later.
  3. In figure 5, scale bar is missing.
  4. In figure 1 and 4, the legends are too blurry to read.
  5. In line 141, should be symbol ‘×’ instead x.

Round 2

Reviewer 2 Report

It was nicely presented towards the findings and all my concerns are addressed accordingly. I’m convinced by the unilateral labyrinthectomy (UL) model using intratympanic injection of ferric chloride (FeCl3), and I think it is full of novelty in unveiling the unbalanced distribution of the directional preference of otolith-related neurons after UL.

This manuscript is a resubmission of an earlier submission. The following is a list of the peer review reports and author responses from that submission.

Round 1

Reviewer 1 Report

Nguyen and colleagues provide a manuscript titled "Neural interruption by unilateral labyrinthectomy biased the directional preference of otolith-related neurons."  The authors describe the directional preference of otolith-related neurons to linear translations after chemical labyrinthectomy.  This work is of interest as understanding the importance of convergent neurons as well as the relationship between labyrinths will improve our understanding of vestibular compensation.

Overall, I find this work well described and thorough.  The authors have established the rationale for the research and the research gap they aim to address.  I have some comments regarding a few areas of clarity.

What is the time course of FeCl3 ototoxicity?  The prior research that I have read on this method conducted evaluation at 3 hours post-injection (Lee et al 2016).  Is there any concern about not achieving maximal labyrinth deficit?  Is there a known effect of FeCl3 on otolith hair cells?  Behaviorally this seems so, but what do the otolith look like histopathologically?  Do you have examples of otolith damage post-FeCl3?

Regarding behavioural testing, were all tests completed on all animals? The manuscript notes that "behavioral tests were selectively applied in the UL models" (pg 4, line 175).  How were these selected per animal or were all animals evaluated using this protocol?

Regarding groups, what was the rationale for having 14 right versus 5 left models?

Because of the group differences, it may be useful to include the percentages in Table 1 for clarity.

Author Response

Please see the attachment, but I don't have a space for the modified manuscript. I uploaded only my responses to your comments in this space. Please let me know if you need to see the modified manuscript.

Reviewer 2 Report

Nguyen et al. investigate the preferred direction sensitivity of otolith target neurons in VN before and after labyrinthectomy. They found that the preferred direction is changed after labyrinthectomy and that this change is more important for the convergent cells that receive canals and otoliths inputs.  

My comments can be summarized in three points:

  • The authors claim that they are recording in VN. They have to provide evidences of it. They could show the response of other neurons types found in VN and/or the response to eye movements. The authors have to demonstrate also that these neurons are Vestibular Only cells.
  • Otolith target neurons in VN are Spatio-Temporal convergent. Their response preferred direction is not binary (ipsi vs contra) but is distributed in 3D (or in 2D if constrains to the horizontal plain). It would have been interesting to test these cells during fore-aft linear accelerations in order to compute a better estimate of the maximal sensitivity vector. At least, it would be worth mentioning in the discussion.
  • Is it surprising that the effect of the labyrinthectomy is stronger in the convergent neurons? Since, the labyrinthectomy affect the canals and the otoliths, it seems trivial to me. Could you further develop your arguments about it? and maybe provide a better discussion of the implications of this study.   

As a minor comment, it is striking that your contra-preferred example neuron has a higher gain and CV than your ipsi-preferred example (figure 3). Is it an exception or is it seen at the population level? It would be very interesting if it is the case.

The study lacks basic controls. They have to provide these controls before the results can be considered. 

Author Response

(The authors gave the same response as above.)
